# Dengue virus reduces *AGPAT1* expression to alter phospholipids and enhance infection in *Aedes aegypti*

Thomas Vial [1,2], Wei-Lian Tan[2], Benjamin Wong Wei Xiang[2], Dorothée Missé[3], Eric Deharo[1], Guillaume Marti[1☯*], Julien Pompon[2,3☯*]

1 UMR 152 PHARMADEV-IRD, Université Paul Sabatier-Toulouse 3, Toulouse, France, 2 Programme in Emerging Infectious Diseases, Duke-NUS Medical School, Singapore, 3 MIVEGEC, IRD, CNRS, Univ. Montpellier, Montpellier, France

☯ These authors contributed equally to this work.
* guillaume.marti@univ-tlse3.fr (GM); julien.pompon@ird.fr (JP)

**Data Availability Statement:** Raw data were deposited in the MassIVE data repository under number MSV000083868; ftp://massive.ucsd.edu/MSV000083868/.

## Abstract

More than half of the world population is at risk of dengue virus (DENV) infection because of the global distribution of its mosquito vectors. DENV is an envelope virus that relies on host lipid membranes for its life-cycle. Here, we characterized how DENV hijacks the mosquito lipidome to identify targets for novel transmission-blocking interventions. To describe metabolic changes throughout the mosquito DENV cycle, we deployed a Liquid chromatography–high resolution mass spectrometry (LC-HRMS) workflow including spectral similarity annotation in cells, midguts and whole mosquitoes at different times post infection. We revealed a major aminophospholipid reconfiguration with an overall early increase, followed by a reduction later in the cycle. We phylogenetically characterized acylglycerolphosphate acyltransferase (AGPAT) enzyme isoforms to identify those that catalyze a rate-limiting step in phospholipid biogenesis, the acylation of lysophosphatidate to phosphatidate. We showed that DENV infection decreased *AGPAT1*, but did not alter *AGPAT2* expression in cells, midguts and mosquitoes. Depletion of either AGPAT1 or AGPAT2 increased aminophospholipids and partially recapitulated DENV-induced reconfiguration before infection *in vitro*. However, only AGPAT1 depletion promoted infection by maintaining high aminophospholipid concentrations. In mosquitoes, AGPAT1 depletion also partially recapitulated DENV-induced aminophospholipid increase before infection and enhanced infection by maintaining high aminophospholipid concentrations. These results indicate that DENV inhibition of *AGPAT1* expression promotes infection by increasing aminophospholipids, as observed in the mosquito's early DENV cycle. Furthermore, in AGPAT1-depleted mosquitoes, we showed that enhanced infection was associated with increased consumption/redirection of aminophospholipids. Our study suggests that DENV regulates aminophospholipids, especially phosphatidylcholine and phosphatidylethanolamine, by inhibiting *AGPAT1* expression to increase aminophospholipid availability for virus multiplication.

**Funding:** This work was supported by a grant from National Medical Research Council, Singapore, (NMRC/ZRRF/0007/2017) awarded to JP, a grant from the Ministry of Education, Singapore, (MOE2015-T3-1-003) partially awarded to JP, and by the Duke-NUS Signature Research Programme funded by the Agency for Science, Technology and Research (A*STAR), Singapore, and the Ministry of Health, Singapore. The funders had no role in study design, data collection and analysis, decision to publish, or preparation of the manuscript.

**Competing interests:** The authors have declared that no competing interests exist.

## Author summary

Dengue is endemic in tropical and subtropical regions, and has now encroached onto temperate regions because of the geographic expansion of its vector, *Aedes aegypti*. In the absence of effective vaccine and curative drug, the sole intervention relies on containment strategies using insecticide. However, occurrence of insecticide resistance diminishes vector control efficacy. Here, we explore the nascent field of mosquito metabolomics as part of our discovery effort for new transmission-blocking targets. Dengue virus (DENV) relies on host metabolome, specifically the lipid membrane to complete its life-cycle. However, little is known about how DENV subverts the mosquito physiology. Using high-resolution mass spectrometry, we described metabolic changes incurred by DENV throughout the mosquito cycle, from cellular replication onset to systemic infection. Membrane phospholipids were highly reconfigured and were associated with reduced expression of *AGPAT1*, an enzyme involved in their biogenesis. AGPAT1 depletion partially recapitulated DENV-induced metabolic reconfiguration and enhanced infection by maintaining high phospholipid concentrations. These phospholipids were then consumed/redirected later in the mosquito DENV cycle. Our work comprehensively describes metabolic changes associated with DENV infection. In addition, we reveal how DENV subdues the lipidome for its benefit by demonstrating the role of phospholipids in mosquito infection.

## Introduction

Increased global distribution of dengue virus (DENV) is driven by the expansion of its mosquito vectors, mainly *Aedes aegypti* [1]. An estimated 400 million infections occur yearly in over 100 countries [2] and cause a range of symptoms from flu-like illness to potentially lethal complication called severe dengue. Without approved antiviral drug, treatment is limited to supportive care. Although dengue vaccine is now licensed in several countries, it has variable efficacy against all dengue serotypes [3], and is only suitable for dengue-seropositive patients [4]. To curb dengue epidemics, containment strategy mainly relies on vector control that includes the use of insecticides [5]. However, insecticide resistance is rapidly developing, compromising the efficacy of the only available intervention [6]. Characterization of viral metabolic requirements in mosquitoes will identify targets for novel chemical-based control strategies [7].

As obligate and intracellular parasites, viruses rely on the host to fulfill their metabolite requirements. Glycolysis, amino acid and lipid pathways provide the energy and structural compounds necessary for multiplication. In human cells, DENV alters energy [8,9], glycolysis [10], nucleic acid [11], mitochondrial [12] and lipid metabolisms [13–15]. As an envelope virus, DENV is particularly dependent on host-derived lipid membranes, with which it interacts for entry, replication, translation, assembly and egress [7,16]. In mosquito cells and midgut, DENV reconfigures the lipid profile as indicated by lipidomics [17,18] and transcriptomics [19], particularly altering the membrane lipids such as phospholipids (PL) and sphingolipids. Chemical inhibition of lipid synthesis confirmed the DENV requirements for lipids in human and mosquito cells [20]. Several mechanisms related to elevated autophagy [13] and recruitment of lipogenesis enzymes to the replication complex [21] have been elucidated in the mammalian host. However, how DENV reconfigures lipids in mosquitoes remains unknown.

PL *de novo* biogenesis is initiated by two types of acyl-transferases that sequentially add two acyls to one glycerol-3-phosphate (G3P) [22,23]. The second addition is catalyzed by 1-acyl-sn-glycerol-3-phosphate O-acyltransferases (AGPAT) that transform lysophospatidate

(lysoPA) in phosphatidate (PA) [24]. PAs are then used to produce all PLs, positioning AGPATs as rate-limiting enzymes of PL biogenesis. PLs are produced within the endoplasmic reticulum and subject to swift reconfiguration to meet the cell needs [25,26]. DENV translation, replication and assembly harness the endoplasmic reticulum membranes, suggesting potential alteration of PL biogenesis.

Following an infectious blood meal, DENV infects the mosquito midgut, multiplies, and propagates to the whole mosquito body, including salivary glands, from where it is expectorated during subsequent blood feeding [27]. Throughout various infected tissues, the virus modifies its metabolic environment. In this study, we aim to understand how DENV modifies mosquitoes' metabolome. Using high-resolution mass spectrometry, we explored the metabolic changes throughout the DENV cycle, in *Ae. aegypti* Aag2 cell line, midguts and whole mosquitoes at different times post infection. Infection-induced metabolic alteration mostly affected PLs and was associated with *AGPAT* expression regulations. Combining metabolomic profiling with RNAi-mediated AGPAT depletions, we partially recapitulated the infection-induced PL reconfiguration and demonstrated its pro-viral impact. Eventually, PL profiling upon infection both *in vitro* and *in vivo* in PL-altered environments indicated that increased PL redirection/consumption favored virus multiplication. Our study reveals how DENV reconfigures the metabolome and identifies PLs as important components in the virus lifecycle.

## Results

### Mosquito phospholipidome is reconfigured throughout DENV infection

To describe the metabolic changes in *Ae. aegypti*, we designed an untargeted multidimensional approach that covered the different stages of DENV cycle (Fig 1A). Extracts from Aag2 cells at 6, 12, 24 and 48 h post infection (hpi) represented changes caused by entry, initiation of replication, replication and virion production (S1A and S1B Fig) [28]. Extracts from midguts at 1 and 7 days post oral infection (dpi) represented infection onset and replication peak, respectively (S1C and S1CD Fig) [27]. Extracts from whole mosquitoes at 1, 7 and 14 dpi represented the different dissemination stages in the mosquito body (S1C and S1CD Fig) [27]. The extracts were analyzed using a Liquid Chromatography-High Resolution Mass Spectrometry (LC-HRMS) metabolomic workflow that detects polar and nonpolar metabolites (S2 Fig). We detected 667, 486 and 1121 compounds in the cells, midgut and whole mosquitoes, respectively (Table 1). Comparison of MS spectra with available non-mosquito databases enabled the annotation of 77% of the peaks. To annotate the remaining mosquito-specific metabolites, we deployed MS spectral similarity network (S3 Fig; S1 Table) [29]. Based on the spectral similarity with database-identified features, we identified the class of 20 MS features without homologues in the databases, increasing the identification coverage to 77, 70 and 82% in the cells, midguts and whole mosquitoes, respectively (Table 1). Among all annotated metabolites, we detected 29% of PLs, 21% of non-PL lipids, 8% of amino acids and peptides, 5% of organic acids, 4% of carbohydrates and 33% of other minor classes (fewer than 3 occurrences). The distribution of metabolite classes indicates that the metabolomic workflow had a broad coverage, although slightly biased towards lipids.

Infection regulated 67, 24 and 181 unique metabolites in cells, midguts and whole mosquitoes, respectively (Fig 1B; S2 Table). The PL class had the highest number of regulated metabolites in the midgut and mosquito, and the third most regulated in cells. Fatty acyls were also differentially regulated in the three tissue levels, while amino acids and carbohydrates were regulated in cells and mosquitoes.

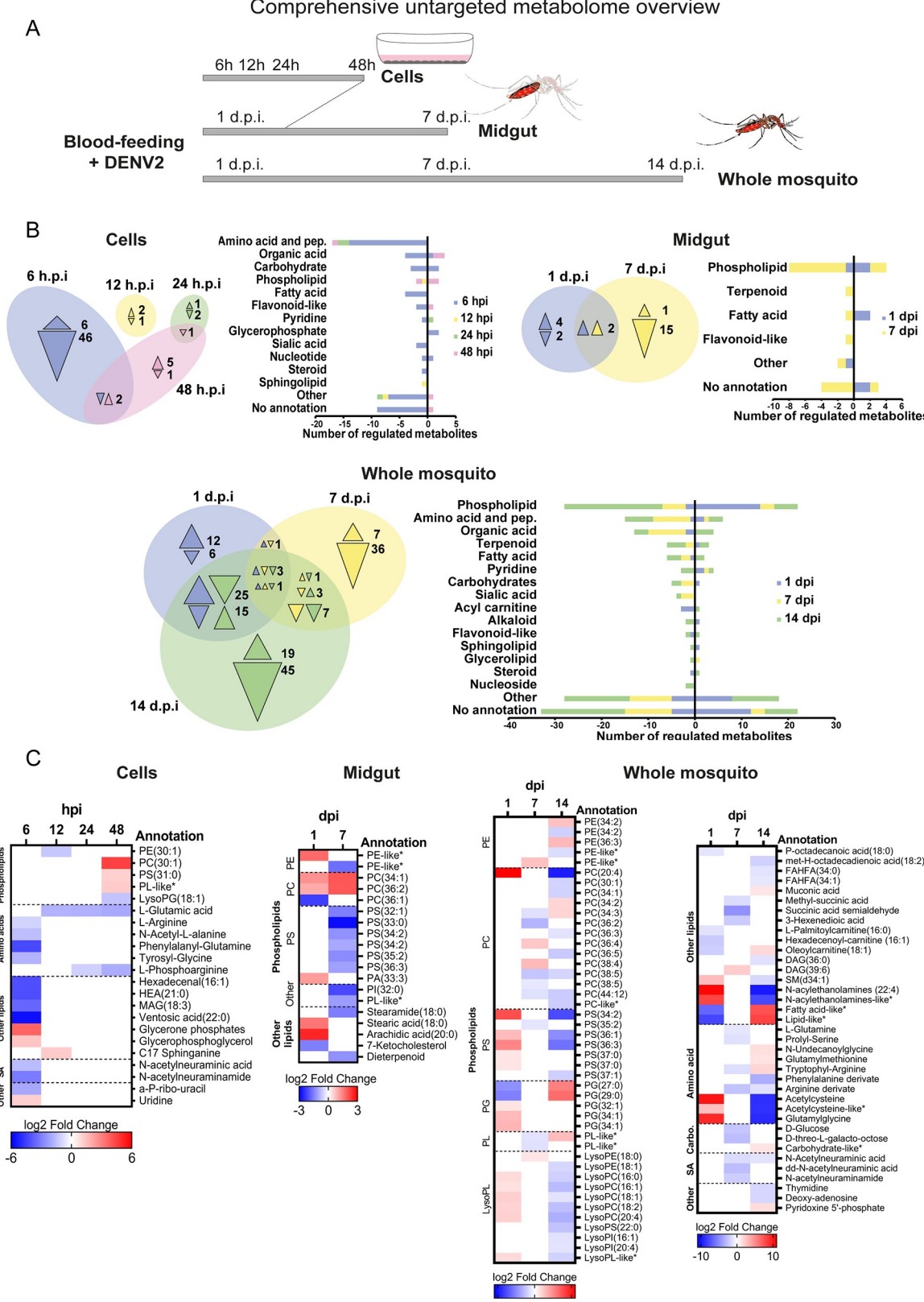

**Fig 1. Aminophospholipid composition is altered throughout DENV infection in mosquitoes.** (A) Schematic of the multidimensional strategy deployed to profile the mosquito metabolome after DENV infection. Aag2 mosquito cells were infected with DENV at an MOI of 5 and collected at 6, 12, 24 and 48 h post infection (hpi). Mosquitoes were orally infected with $10^7$ pfu/ml of DENV. Midguts and whole mosquitoes were collected at 1, 7 and 14 days post infection (dpi). Metabolic extracts were analyzed by LC-HRMS as in S2 Fig. (B) Venn diagrams show regulated metabolites at the different time points and bars indicate class distribution within tissue levels. Triangles indicate direction of regulation and are color-coded with regards to the time point. Uninfected condition was used as control. (C) Fold changes of annotated and significantly regulated metabolites ($|\log_2$ fold change| > 1 and p-value < 0.05) by DENV in cells, midguts and mosquitoes as compared to mock infection. Only metabolites from the general metabolism (i.e., lipid, carbohydrate, amino acid and peptide, nucleotide and nucleoside, sialic acid) are shown. *, indicates metabolite annotations determined by spectral similarity. Carbo., carbohydrate; SA, sialic acid; PL, phospholipid; PE, phosphatidylethanolamine; PC, phosphatidylcholine; PS, phosphatidylserine; PA, phosphatidic acid; PI, phosphatidylinositol; PG, phosphatidylglycerol; LysoPS, lysophosphatidylserine; LysoPC, lysophosphatidylcholine; LysoPE, lysophosphatidylethanolamine; LysoPG, lysophosphatidylglycerol; LysoPI, lysophosphatidylinositol; SM, Sphingomyelin; DAG, Diacylglycerol; MAG, Monoacylglycerol; FAHFA, Fatty Acid ester of Hydroxyl Fatty Acid; NAE, N-acylethanolamine; HEA, Heneicosanoic acid; pep., peptides.

Several classes of PLs were regulated in all three tissues (Fig 1C, S4 Fig). Aminophospholipids (aminoPL) are the major constituents of membranes and are synthesized in the endoplasmic reticulum [30]. AminoPLs include phosphatidylethanolamine (PE), phosphatidylcholine (PC) and phosphatidylserine (PS) [31]. Although infection altered different species of aminoPLs in the different tissue-time combinations, we observed a general increase at the beginning of DENV cycle, followed by a reduction at the end. In cells, the majority of regulated aminoPLs (3 out of 5) were upregulated at 48 hpi. In midguts and mosquitoes a total of 9 aminoPLs were increased at 1 dpi, whereas 15 and 14 were reduced at 7 and 14 dpi, respectively.

Phosphatidylglycerols (PG) are another group of PL. Although PGs have a lower abundance than aminoPLs, they are also constituents of membranes and synthesized in the mitochondria [32]. Although only detected in mosquito extracts, PGs responded differently to infection when compared with aminoPLs (Fig 1C). Two shorter PGs decreased at 1 dpi and increased at 14 dpi, and three longer PGs were upregulated at 1 dpi.

Lysophopholipids (lysoPL) are produced from fatty acid remodeling of the different PL classes [33]. In cells, one lysoPG was downregulated at 48 hpi (Fig 1C). In mosquitoes, the lysoPCs followed the same trend as aminoPLs, with 6 increasing at 1 dpi and decreasing at 14 dpi.

Different types of PL precursors were regulated. PA, the direct precursor of all PLs [24,31], was upregulated at 1 dpi in midgut, similar to aminoPLs. Two diacylglycerols (DAG), intermediates for PE and PC productions, were up and downregulated in mosquitoes at 7 and 14 dpi, respectively. Fatty acyls can be incorporated into a glycerol head to produce PL and were regulated. Fatty acyls with an ethanolamine group can be related to PE either via degradation or as a precursor. Similar to aminoPLs, two acyl-ethanolamine increased at 1 dpi and decreased at 14 dpi in mosquitoes. In accordance with previous studies [34], one sphingomyelin was regulated in mosquitoes and one cholesterol compound was depleted in midgut.

Our comprehensive metabolomic profiling revealed that DENV infection profoundly reconfigures the phospholidome. The aminoPLs including PE, PC and PS increased at the

**Table 1. Summary of metabolites detected across mosquito tissues.** ¶MS Finder (HMDB, ChEBI, LipidMAPS, LipidBlast) score annotation ≥ 5; * p-value < 0.05 as indicated by unpaired t-test and $|\log_2$ Fold Change| ≥ 1.

| Tissue | Aag2 cells | | | | A. aegypti Midgut | | A. aegypti Mosquito | | |
|---|---|---|---|---|---|---|---|---|---|
| Unique peak detected with MS/MS | 667 | | | | 486 | | 1121 | | |
| Annotated peaks (%)¶ | 77% | | | | 70% | | 82% | | |
| Metabolites significantly regulated* | 6 hpi | 12 hpi | 24 hpi | 48 hpi | 1 dpi | 7 dpi | 1 dpi | 7 dpi | 14 dpi |
| Total | 54 | 3 | 4 | 9 | 8 | 18 | 63 | 59 | 119 |
| Annotated (%) | 83% | 100% | 100% | 89% | 75% | 72% | 73% | 78% | 79% |

beginning of DENV cycle and decreased later on. A large phospholipidome reconfiguration was previously observed in DENV-infected midguts at 3, 7 and 11 dpi [18]. However, in this previous kinetic study, PLs were mostly up-regulated throughout the infection cycle. They reported a decrease in lysoPL abundance, which we observed at 14 dpi in mosquitoes. The authors also noted an increase in sphingolipids, which we saw in mosquitoes but only for one species. Those variations between the studies can stem from methodological differences in extraction and analyses, or from biological differences in mosquito colony and virus strain. Taken together, our results and those of others indicate that DENV infection reconfigures lipid membrane composition.

## DENV infection modulates expression of *AGPAT1* that is involved in PL biogenesis

To determine how PLs are reorganized upon DENV infection, we first characterized the rate-limiting AGPAT enzymes in *Ae. aegypti* (Fig 2A). In humans, there are five AGPAT isoforms with different activities depending on four motif sequences [35]. Motifs I and IV bind to acyl-CoA and catalyze lysoPA to PA acylation, while motifs II and III bind to lysoPA. Human AGPAT (hAG-PAT) 1 and 2 are localized in the endoplasmic reticulum and have the highest acyltransferase activity and lysoPA affinity [36]. hAGPAT3-5 have lower transferase activity and target different substrates that lysoPA such as lysoPL, participating in PL remodeling [37]. *Aedes aegypti* also has five AGPAT isoforms: AGPAT1 (AAEL011898), AGPAT2 (AAEL001000), AGPAT3 (AAEL011902), AGPAT4 (AAEL014026) and AGPAT5 (AAEL011901). Based on amino acid similarity, AGPAT1, 2, 3 and 5 cluster with hAGPAT1 and 2, whereas AGPAT4 clusters with hAGPAT3-5 (Fig 2B; S5 Table). Further classification based on functional motifs that are define the biochemical activity [38] establishes AGPAT1, 2, 3 as homologues of hAGPAT1 and 2 (Table 2), suggesting that they all share the acyltransferase activity and lysoPA affinity.

To test whether infection-induced phospholipidome reconfiguration is associated with AGAPT regulation, we quantified *AGPAT1* and *2* expressions. Interestingly, we observed that *AGPAT1* was downregulated at 24 and 48 hpi in cells, at 1 and 7 dpi in midguts and at 1, 7 and 14 dpi in mosquitoes (Fig 2C–2E). In contrast, *AGPAT2* was not significantly regulated in cells, midguts and mosquitoes (Fig 2C–2E). This partially corroborates a transcriptomic study [39] that shows *AGPAT1* downregulation and *AGPAT2* upregulation at 1, 2 and 7 days post DENV-inoculation in mosquitoes (S5 Fig). To test whether virus replication was required for *AGPAT1* down-regulation, we incubated cells with UV-inactivated virus (S6A Fig). *AGAPT1* expression did not vary between mock and UV-inactivated DENV at 24 and 48 hpi (S6B Fig), indicating that active infection is required.

Altogether, our results and those of others suggest that *AGPAT1* down-regulation correlates with DENV phospholipidome reconfiguration.

## Depletion of AGPAT1 but not AGPAT2 promotes DENV infection by increasing aminoPL concentrations in cells

Based on the association between the infection-induced phospholipidome reconfiguration and *AGPAT1* down-regulation, we hypothesized that AGPAT1 mediates the phospholipidome reconfiguration that promotes DENV infection. First, we described how both AGPAT1 and 2, the latter used as control, regulate PL biogenesis in non-infected mosquito cells. LC-HRMS polar mode detection was used to target phospholipid metabolites (S2 Fig). Depletion of either AGPAT1 or 2 increased the concentrations of aminoPLs (Fig 3A–3C; S6 Table), as previously observed in human cells [40]. Strikingly, PC (34:1), PC (38:5) and G3P were upregulated by both AGPAT1 and 2 depletion. AGPAT2 depletion also increased two other PCs and

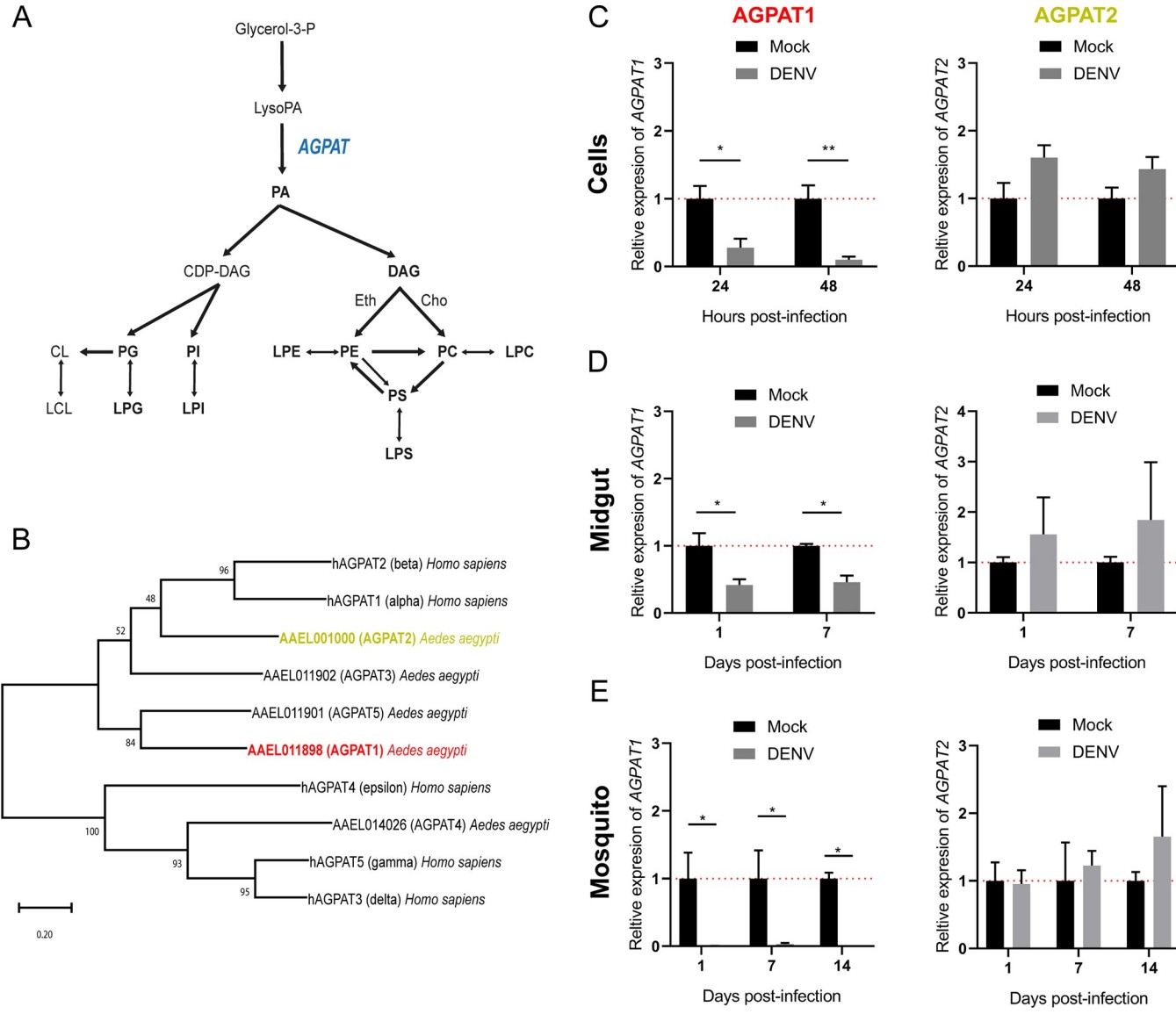

**Fig 2. DENV infection decreases *AGPAT1* but not *AGPAT2* expression.** (A) An overview of phospholipid biogenesis. Sequential additions of two acyl-coA to one G3P produce a LysoPA and a PA. Several AGPATs mediate the second addition. PA forms either DAG or CDP-DAG, each generating a different set of phospholipids. DAG produces aminophospholipids by addition of a Cho or Eth group. Aminophospholipids are also produced by acylation of lysoPL and base group modifications. CDP-DAG produces PI and PG, the latter being transformed in CL by combination with a second PG. Phospholipases cleave off an acyl chain from PL to produce lysoPL. G3P, glycerol-3-phosphate; LysoPA, lysophosphatidate; PA, phosphatidate; AGPAT, acyl-sn-glycerol-3-phosphate acyltransferases; DAG, diacylglycerol; CDP-DAG, cytidine diphosphate diacylglycerol; PC, phosphatidylcholine; LPC, lysophosphatidylcholine; Cho, choline; PE, phosphatidylethanolamine; LPE, lysophosphatidylethanolamine; Eth, ethanolamine; PS, phosphatidylserine; LPS, lysophosphatidylserine; PI, phosphatidylinositol; LPI, lysophosphatidylinositol; PG, phosphatidylglycerol; LPG, lysophosphatidylglycerol; CL, cardiolipin; LCL, lysocardiolipin. (B) Maximum likelihood tree between AGPATs from *Ae. aegypti* and humans. (C) *AGPAT1* and *AGPAT2* expressions in DENV-infected cells at 24 and 48 hpi with an MOI of 5. Expressions of *AGPAT1* and *AGPAT2* (D) in midguts at 1, 7 dpi and (E) in whole mosquitoes at 1, 7 and 14 dpi with DENV at $10^7$ pfu/ml. (C-E) *Actin* expression was used for normalization. Bars show means ± s.e.m from 4 independent wells or 3 pools of 5 midguts or 5 mosquitoes. *, p-value < 0.05; **, p-value<0.01; as indicated by unpaired t-test.

decreased two forms of sphinganine, a precursor of sphingolipid biosynthesis [41] (Fig 3D; S7A Fig), and six amino acids or nucleoside (arginine, cysteine, methionine, glutamic acid, oxidized glutathione and adenosine) (S6 Table). Of note, depletion of one of the AGPATs did not alter expression of the other (S8 Fig), indicating the enzyme-specificity of the aminoPL

**Table 2. Acyltransferase motif comparison between human and *Ae. aegypti* AGPAT homologues.** Purple indicates amino acid residues highly conserved across all hAGPATs and yellow in hAGPAT1 and 2 only.

| Specie | Protein | RefSeq | Amino acids | Motif I acyl-CoA binding and catalysis | Motif II LPA binding | Motif III LPA binding | Motif IV acyl-CoA binding and catalysis |
|---|---|---|---|---|---|---|---|
| *Homo sapiens* | hAGPAT1 (alpha) | NP_006402 | 283 | VSN H QS SL D LL G M | A GV I F I D R K R | V FPEGT RN H | VPIV P I V M SS |
| *Homo sapiens* | hAGPAT2 (beta) | NP_006403 | 278 | VSN H QS IL D MM G L | G GV FF I N R Q R | IY PEGT RN D | VPIV PV V Y SS |
| *Homo sapiens* | hAGPAT3 (gamma) | NP_064517 | 376 | IL N H NFEI D FLCG | LEIV F CK R KW | LYC EGT R FT | YHLL P RTKGF |
| *Homo sapiens* | hAGPAT4 (delta) | NP_064518 | 378 | V L N H KFEI D FLCG | TEMV F CS R KW | IHC EGT R FT | HHLL P RTKGF |
| *Homo sapiens* | hAGPAT5 (epsilon) | NP_060831 | 364 | LA N H QS TV D WIVA | QHGGIYVKRS | I FPEGT R YN | HVLT P RIKAT |
| *Ae. aegypti* | AGPAT1 (AAEL011898) | EAT35978 | 273 | LM N H QS A L D LVVL | WGTL F I N R KN | F FPEGT R GD | GYIQ PV V I S K |
| *Ae. aegypti* | AGPAT2 (AAEL001000) | EAT47921 | 398 | V A N H QS SL D ILG M | S G LI F I D R KN | V FPEGT R RN | L PI M PV V Y SS |
| *Ae. aegypti* | AGPAT3 (AAEL011902) | EAT35981 | 308 | MA N H QS SM D ILG L | A G IT F I N R KN | IY PEGT R FP | VPI I PV V F S H |
| *Ae. aegypti* | AGPAT4 (AAEL014026) | EAT33698 | 387 | LM N H TYEV D WLVG | AEFV F LE R SF | LNA EGT R FT | HHLI P RTKGF |
| *Ae. aegypti* | AGPAT5 (AAEL011901) | EAT35980 | 280 | LI N H QS AI D IVML | V GV V F I D R KN | I FPEGT R HD | SI I QSIIV S K |

alterations. These results confirm the roles of AGPAT1 and 2 in PL biogenesis, and reveal differences between the two enzymes.

To test whether AGPAT-mediated reconfiguration of aminoPLs promotes DENV infection, we quantified infection in cells depleted of either AGPAT1 or 2. While DENV titer was not altered by AGPAT2 depletion, it increased 2.09 ± 0.36 fold (p-value = 0.0012) following AGPAT1 depletion (Fig 3E). Next, we described how DENV infection altered the lipidome in cells depleted of either AGPAT1 or 2. Infection in AGPAT1-depleted cells, but not in AGPAT2-depleted cells, mostly increased aminoPLs as compared to infected wild-type cells (Fig 3C, S7 Table). Specifically, two PEs, two PCs, two PSs, and one lysoPE were upregulated (Fig 3F; S7B Fig). These results indicate that: (i) depletion of AGPAT1 prior infection amplifies the aminoPL increase observed in early mosquito DENV infection (Fig 1C), and (ii) AGPAT1-mediated aminoPL reconfiguration is associated with increased DENV production. Therefore, *AGPAT1* downregulation by infection generates a pro-viral environment.

To test whether AGPAT1 effect on DENV was related to aminoPLs, we modified the AGPAT1-induced reconfiguration of aminoPLs by media supplementation. In PL biogenesis, PA produces DAG, which is transformed into PE by addition of ethanolamine (Fig 2A). Extracellular source of ethanolamine influences phospholipid metabolism [42], especially PEs that were altered by AGPAT1 depletion (Fig 3F). Therefore, we measured the impact on DENV gRNA of ethanolamine supplementation upon AGPAT1 depletion. Controls were non-depleted cells without supplementation (standard media that does not contain ethanolamine), non-depleted cells with ethanolamine supplementation and AGPAT1-depleted cells without supplementation. While ethanolamine supplementation did not alter gRNA in non-depleted cells, the increase observed upon AGPAT1-depletion was reverted to non-depleted non-supplemented control when ethanolamine was supplemented (S9 Fig). These results confirm the role of AminoPLs in AGPAT1 increase of DENV multiplication and incriminate metabolites downstream of DAG as important for DENV.

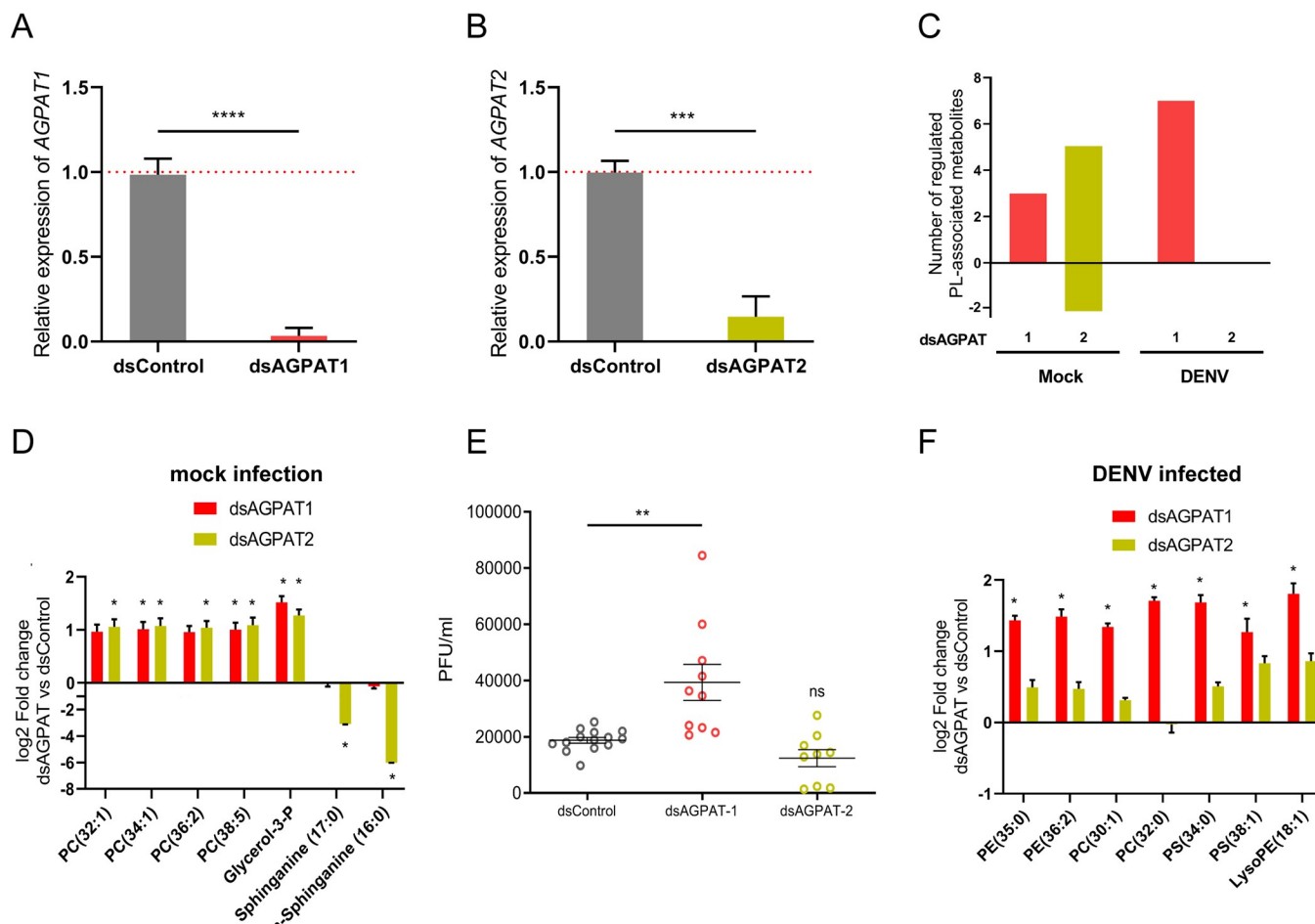

**Fig 3. AGPAT1 but not AGPAT2 depletion increases DENV multiplication and aminoPL in cells.** Aag2 cells were transfected with either dsRNA against *AGPAT1* or *2* (dsAGPAT1 or 2) or with dsRNA control (dsControl). At 24 h post transfection, cells were infected with DENV at MOI of 1 or mock infected. Supernatant was collected at 48 hpi. Expressions of (A) *AGPAT1* and (B) *AGPAT2* in mock-infected cells at 72 h post transfection. *Actin* expression was used for normalization. Bars show mean ± s.e.m. (C) Number of phospholipid-related metabolites significantly regulated. (D) Impact of AGPAT1 or 2 depletion on the lipidome of mock-infected cells at 72 h post transfection. *, p-value <0.05 and |log$_2$ fold change| > 1. (E) Impact of AGPAT1 or 2 depletion on DENV production at 48 hpi as determined by plaque forming unit (pfu) assay. Bars show mean ± s.e.m. and each point represents independent wells. (F) Impact of AGPAT1 or 2 depletion on the lipidome of infected cells at 48 hpi. *, p-value < 0.05 and |log$_2$ fold change| > 1. (A, B, D and F) result from three biological replicates. (A, B) ***; p-value < 0.001; ****, p-value < 0.0001, as indicated by unpaired t-test. (E) **, p-value < 0.01, as indicated by Dunnett's test. PE, phosphatidylethanolamine; PC, phosphatidylcholine; PS, phosphatidylserine; LPE, lysophosphatidylethanolamine.

## AGPAT1 depletion promotes DENV infection by amplifying aminoPL reconfiguration in mosquitoes

To further characterize how AGPAT1-mediated PL alteration increases DENV infection, we depleted AGPAT1 in mosquitoes (Fig 4A). We first described how AGAPT1 regulates PLs in mosquitoes that fed on a non-infectious blood meal (Fig 4B; S10 Fig; S8 Table). As in cells, AGPAT1 mostly regulated aminoPLs and their derivatives (S8 Table). At 2 days post oral feeding, one DAG and one PA, two key precursors of PLs, and one PS were increased. At 7 days post oral feeding, eight lysoPLs (one lysoPI, two lysoPGs, one lysoPS, two lysoPEs and two lysoPCs) were downregulated, similarly to those in DENV-infected wild-type mosquitoes at 14 dpi (Fig 1C). Strikingly, some metabolites were similarly regulated by either AGPAT1-depletion or DENV infection. PS (34:2) was upregulated by either AGPAT1-depletion at 2 days post oral feeding or DENV infection at 1 dpi. LysoPI (20:4), lysoPE (18:1), lysoPC (16:1) and lysoPC

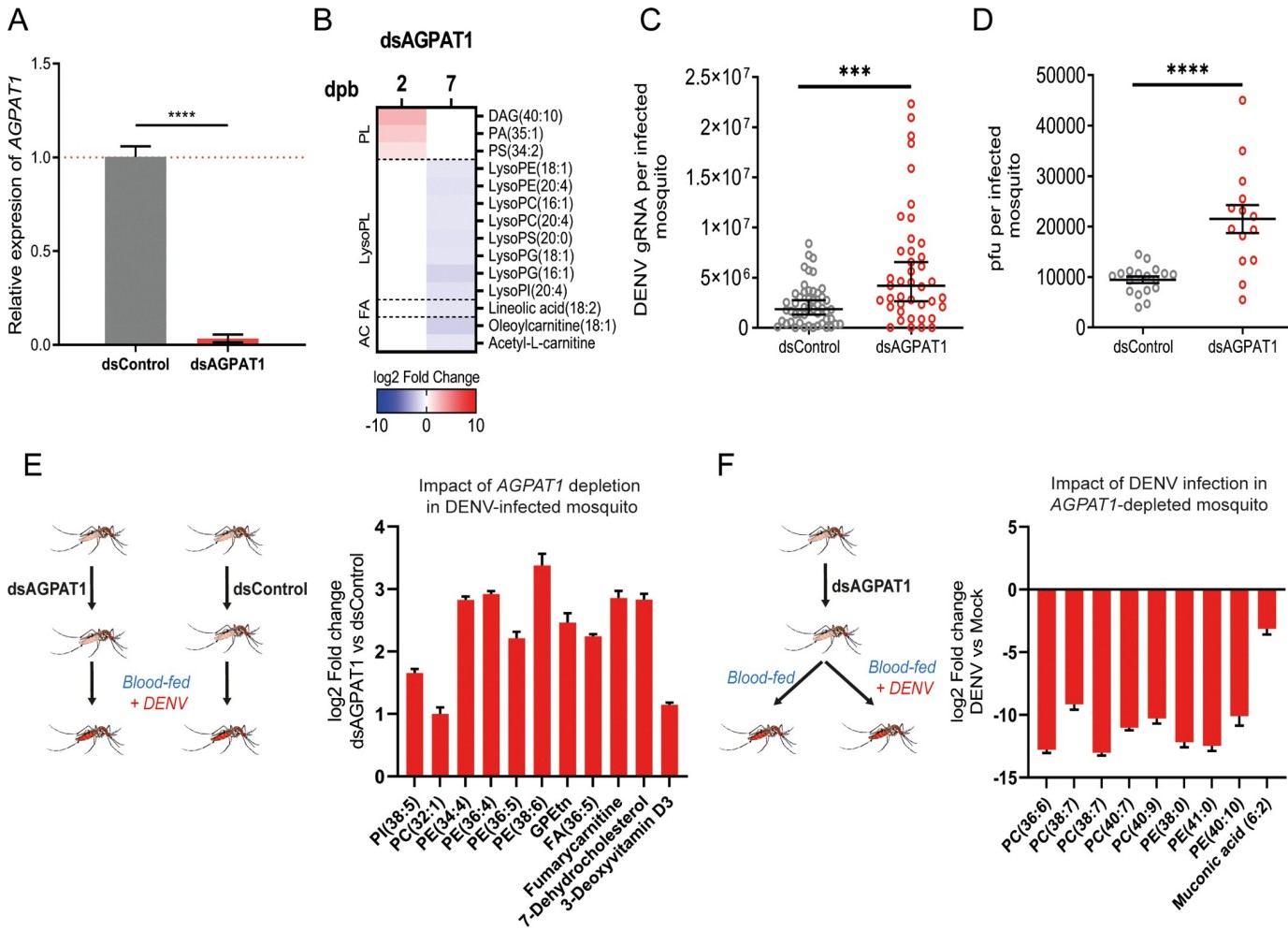

**Fig 4. AGPAT1 depletion increases DENV multiplication and consumption of aminoPLs in mosquitoes.** Mosquitoes were injected with either dsRNA against *AGPAT1* (dsAGPAT1) or dsRNA control (dsControl). Two days later, mosquitoes were orally fed with non-infectious blood or infectious blood containing DENV at 10⁷ pfu/ml. (A) Validation of *AGPAT1* silencing in non-infected mosquitoes at two days post dsRNA injection. *Actin* expression was used for normalization. Bars show mean ± s.e.m. from 3 pools of 5 mosquitoes each. ****, p-value < 0.0001 as indicated by unpaired t-test. (B) Impact of AGPAT1 depletion on the lipidome of mosquitoes at 2 and 7 days post non-infectious blood feeding (dpb). Impact of AGPAT1 depletion on (C) DENV gRNA copies and (D) viral load measured as pfu/ml at 7 days post oral infection (dpi). Bars indicate mean (C) or geometric means ± 95% CI (D) with each dot representing one mosquito. ***, p-value < 0.001 ****, p-value < 0.0001 as indicated by unpaired t-test (C) or Mann-Whitney test (D). (E) Impact of AGPAT1 depletion on the lipidome of infected mosquitoes at 7 dpi. (F) Impact of infection on the lipidome of AGPAT1-depleted mosquitoes at 7 dpi. (B, E, F) Only regulated metabolites are shown (p-value < 0.05 and |log₂ fold change| > 1). PL, phospholipid; FA, fatty acyl; AC, acylcarnitine; DAG, diacylglycerol; PA, phosphatidic acid; PS, phosphatidylserine; LPE, lysophosphatidylethanolamine; LPC, lysophosphatidylcholine; LPS, lysophosphatidylserine; LPG, lysophosphatidylglycerol; LPI, lysophosphatidylinositol; PE, phosphatidylethanolamine; PC, phosphatidylcholine; PI, phosphatidylinositol; GPEtn, glycerophosphoethanolamine.

(20:4) were reduced by either AGPAT1-depletion at 7 days post oral feeding or DENV infection at 14 dpi. At 7 days post oral feeding in AGPAT1-depleted mosquitoes, one fatty acyl, two acylcarnitines, oleoylcarnitine (18:1) and acetyl-L-carnitine, were decreased. Of note, one of the regulated acylcarnitines (i.e., oleoylcarnitine) was inversely regulated by DENV infection at 14 dpi (Fig 1C). These results show that AGPAT1 depletion partially reproduces the aminoPL reconfiguration caused by DENV infection.

Next, we found that AGPAT1-depletion increased DENV gRNA (p-value = 0.0004) and titer (p-value < 0.0001) at 7 dpi in whole mosquito (Fig 4C and 4D), confirming that AGPAT1 depletion induces a pro-viral environment *in vivo*. However, the pro-viral impact of AGPAT1 depletion was not observed on infection rate (S11A Fig) and at 2 dpi on gRNA (S11B Fig). We

repeated the experiment using a lower inoculum for oral infection and similarly observed no effect at 2 dpi but a moderate increase of gRNA (p = 0.0421) at 7 dpi (S11C–S11E Fig). It is intriguing that AGPAT1 depletion only increased infection at 7 dpi. This may indicate that DENV infection is necessary to amplify AGPAT1-mediated alteration in establishing a proviral environment. To characterize the AGPAT1-induced proviral environment, we compared the lipidome of infected mosquitoes that were AGPAT1 depleted or not. AGPAT1-depletion increased aminoPL concentrations at 7 dpi (Fig 4E; S9 Table). Specifically, one PC, four PEs and one glycerophosphoethanolamine were upregulated. By artificially modulating the metabolome, we revealed that AGPAT1 depletion favors DENV by increasing aminoPL concentrations.

When profiling metabolic changes throughout mosquito DENV cycle, we observed that aminoPLs were reduced at the end of DENV cycle (Fig 1C). To test whether this reduction occurred when aminoPL concentrations were increased by AGPAT1 depletion (Fig 4E), we examined how DENV modifies the lipidome in AGPAT1-depleted mosquitoes. Control mosquitoes were depleted of AGPAT1 and fed a non-infectious blood meal. At 7 dpi, aminoPLs were drastically decreased by DENV despite AGPAT1-depletion enhancement (Fig 4F; S10 Table). Specifically, three PEs and five PCs were reduced more than 100 folds. These results suggest that, by reproducing infection-induced aminoPL reconfiguration, AGPAT1 depletion amplifies the DENV reconfiguration, thereby promoting infection. Altogether, our results indicate that DENV inhibits *AGPAT1* expression to increase the amplitude of aminoPLs consumption/redirection for virus multiplication.

## Discussion

By combining observational and manipulative approaches *in vitro* and *in vivo*, we deciphered how DENV hijacks the mosquitoes' metabolome. We deployed metabolomic profiling throughout the mosquito DENV cycle, and revealed an overall increase in aminoPLs, followed by a reduction at the end of the cycle. We next showed that aminoPL reconfiguration is partially mediated by infection-induced *AGPAT1* down-regulation, which increases aminoPL concentrations. Because AGPAT1 depletion promotes virus multiplication, our study discovers a mechanism whereby DENV reconfigures the metabolome to its benefit. Furthermore, we show that in an environment richer in aminoPLs (AGPAT1 depletion), infection reduces aminoPL concentrations at a higher and earlier rate than in a wild-type organism, while increasing virus production. This suggests that DENV consumes/redirects aminoPLs, although this is based on correlations between indirect lipidome alteration and virus multiplication. Altogether, we propose a model whereby DENV regulates aminoPL enzymes to increase aminoPL concentrations early during the cycle and consumes/redirects them for its multiplication.

We revealed that DENV-induced phospholipidome reconfiguration is partially mediated by lowering *AGPAT1* expression. Indeed, *AGPAT1* was downregulated upon infection and its depletion partially recapitulated DENV-induced aminoPL reconfiguration both *in vitro* and *in vivo*. That we only partially recapitulated the DENV-induced phospholipidome reconfiguration suggests that other enzymes, such as those directly producing and those hydrolyzing aminophospholipids [43], play a role. AGPAT enzymes can influence PL composition at several levels. By catalyzing the second acylation that produces PA (Fig 2A), AGPAT enzymes are responsible for a rate-limiting step in PL biogenesis [35]. However, we did not observe PA reduction upon AGPAT1 depletion and instead reported an alteration of aminoPLs. This pattern was previously observed in hAGPAT-depleted human cells [40] and was attributed to compensation by other AGPAT isoforms. Lack of AGPAT1 may unbalance substrate competition with other isoforms with different lysoPA and acyl affinity [44]. Consequently, this can

influence PA structure and subsequent PL composition. In addition, certain AGPAT isoforms can transfer acyls to lysoPL, thus, participating in PL remodeling independently of *de novo* synthesis [45]. In mosquito cells, AGPAT2 depletion also altered PL profile. However, *AGPAT2* expression is not reduced by DENV infection and its depletion does not impact virus multiplication. The correlation between expression of *AGPAT* isoforms and their impacts on infection indicates a fine regulation of metabolism by DENV, and identifies a new mechanism by which the virus hijacks the host phospholipidome.

The expression of *AGPATs* is regulated either directly through their transcription factors or indirectly by altering the endoplasmic reticulum topology. Sterol regulatory element binding proteins (SREBP) coordinate fatty acid, sterol and phospholipid metabolisms by transcription-ally regulating enzymes, such as acyl transferases [46]. DENV protein interaction with SREBP, as described with other transcription factors [47], could alter *AGPAT* expressions. In support of this, chemical inhibition of SREBP blocks DENV replication [48]. Alternatively, AGPAT activity is influenced by substrate accessibility, which depends on membrane topology [36]. Several DENV proteins are embedded in the endoplasmic reticulum membrane and modify its topology [49,50]. This can alter endoplasmic reticulum-located AGPAT activity, PL profile and activate acyl-transferase expressions to restore homeostasis [51].

DENV intricately interacts with aminoPL-containing membranes and a change as we report can alter several stages of its life-cycle, such as entry, replication, translation, assembly and egress [7,16]. DENV interacts with the plasma membrane for entry and fusion, and with the endoplasmic reticulum membrane for translation, replication and assembly [28]. These membranes are mostly composed of PE and PC, and PS in lower proportions [52]. Their fluid-ity and topology are determined by the aminoPL structure and concentration. The cylindrical shape of PCs stabilizes lipid bilayers, whereas the conical shape of PEs induces curvature. Sev-eral flavivirus-aminoPL interactions have been described in human cells. PLs present in the DENV envelope are recognized by cellular ligands and mediate entry [53,54]. Replication of flaviviruses induce the invagination of the endoplasmic reticulum by altering PE [55,56] and lysoPC compositions [43]. Alternatively, *AGPAT* expressions could alter anti-viral immune response [57]. In mammalian cells, *AGPAT* overexpressions amplify cellular signaling of cyto-kine [58] that can reduce DENV infection in mosquito cells [59]. Although the precise func-tion of AGPAT1-regulated aminoPLs is unknown and may be multifactorial, previous studies and ours indicate that membrane aminoPL reconfiguration influences DENV multiplication.

In conclusion, our study determines the importance of aminoPL reconfiguration for DENV infection. We also reveal the underlying mechanism for viral phospholipidome recon-figuration. The intricate metabolic interactions between DENV and mosquitoes represents a target to control transmission.

## Materials and methods

### *Aedes aegypti* mosquitoes and cell line

The *Aedes aegypti* colony was established in 2010 from Singapore and was reared at 28˚C and 60% relative humidity with 12h:12h light:dark cycle. Eggs hatched in milliQ water were fed with a mix of fish food (TetraMin fish flakes), yeast and liver powder (MP Biomedicals). Adults were held in rearing cages (Bioquip) supplemented with water and 10% sucrose solution. *Aedes aegypti* Aag2 cells [60] were grown in RPMI-1640 medium (Gibco) with 10% filtered fetal bovine serum (FBS) (Hyclone) and 1% Penicillin-Streptomycin (Gibco). For media sup-plementation, 2 mM of ethanolamine (Sigma) was added to Aag2 growth medium. Cells were maintained in vented culture flasks in a humidified incubator with 5% $CO_2$ at 28˚C. BHK-21

(baby hamster kidney) (ATCC CCL-10) cells were grown in the same media and maintained at 37˚C with 5% $CO_2$.

## Dengue virus

Dengue virus serotype-2 strain ST (DENV) was collected from the Singapore General Hospital in 1997 [61]. DENV was propagated alternatively in Vero (ATCC CCL-81) and C6/36 (ATCC CRL-1660) cells. Virus titer was determined by plaque assay using BHK-21 cells. DENV supernatant was exposed to UV light from biosafety cabinet (Sterilgard III advance, the Baker Company) for 1h at room temperature for inactivation.

## Oral infection of mosquitoes

Two- to four-day-old adult female mosquitoes were starved for 24h before oral feeding on a blood meal containing 40% volume of washed erythrocytes from SPF pig's blood (PWG Genetics, Singapore), 5% of 100 mM ATP (Thermo Fisher Scientific), 5% of human serum (Sigma-Aldrich) and 50% of DENV-2 in RPMI media (Gibco). The virus titer in the blood meal was 2 x $10^7$ pfu/ml and validated by plaque assay. Blood was maintained at 37˚C using hemotek membrane feeder system (Discovery Workshops) with sausage casing for 1.5 h. A control group was allowed to feed on the same mix of SPF pig blood meal without virus. Engorged mosquitoes were visually selected and maintained at 28˚C with water and 10% sucrose solution.

## Cell inoculation

5 x $10^6$ cells were inoculated with DENV at an MOI of 5 in serum-free RPMI media for 1h. The inoculum was then replaced with 2% FBS RPMI media. Mock infection was used as negative control.

## Metabolite extraction from mosquitoes, midguts and cells

At 1, 7 and 14 days post-oral feeding, 10 mosquitoes in 500 µl of ice-cold methanol and water ratio of 80:20 (LCMS grade, Thermo Fisher) were homogenized with bead Mill homogenizer (FastPrep-24, MP Biomedicals) and sonicated for 15 min in an ultrasonic bath (J.R. Selecta) at 4˚C. Homogenates were centrifuged at 10,000 rpm for 1 min at 4˚C to collect 400 µl supernatant. Pellets were further extracted twice by addition of 500 µl of methanol:water (80:20), followed by centrifugation. Supernatants were combined and vacuum-dried (Speed-Vac, Thermo-Scientific) before storage at -20˚C. At 1 and 7 days post-oral feeding, 10 midguts were homogenized in 200 µl of the methanol:water (80:20) solution and 120 µl of supernatant was collected three times with the same protocol. At 6, 12, 24 and 48 h post-inoculation, cells were washed with room temperature 0.9% NaCl and collected in 2 ml of ice-cold methanol:water (80:20) by scraping. Cells were homogenized by ultrasound and extracted thrice as detailed above by adding 500 µl of ice-cold methanol:water (80:20). Three biological replicates were conducted per condition.

## LC-HRMS metabolism profiling

Dry extracts were normalized at 2 mg/mL in methanol:water 80:20 solution and metabolites were detected using two methods. Compounds from medium range polarity to lipophilic substances were detected using a UPLC-UV-QTOF-MS$^E$ instrument (Xevo G2 QTof, Waters) mounted on an electrospray ionization (ESI) source, with a UPLC BEH C18 Acquity column (100 × 2.1 mm i.d., 1.7 µm, Waters) equipped with a guard column. Mobile phase A was of

0.1% formic acid in water, B was 0.1% formic acid in acetonitrile. The flow rate was 400 μl/min and the gradient ran with 98% A for 0.5 min to 20% B over 3.5 min, 98% B for 8 min, held at 98% B for 3 min, and returned in 0.5 min to initial conditions (98% A), finally held for 3.5 min to assure equilibration before the subsequent analysis. Detection was performed at 254 nm by TOF-MS in both electrospray (ESI) negative mode with voltage at 2.5 kV and positive mode with voltage at 3.0 kV. The m/z range was 100–1200 Da with a scan time of 0.1 s. All detected ions were fragmented using MS$^E$ scan with an energy collision ramp from 20 to 50 eV. All analyses were acquired using leucine enkephalin as the lock mass at a concentration of 400 pg/ μl and flow rate 7 μl/min. The injection volume was 2 μl and samples were kept at 10˚C during the whole analysis.

Polar metabolites were profiled using a UPLC-LTQ Orbitrap XL instrument (Ultimate 3000, Thermo Fisher Scientific, Hemel Hempstead, UK) set at 15,000 resolution, with a Zic-pHilic column (150 × 2.1 mm i.d., 5 μm, SeQuant, Merck). Mobile phase A was 20 mM ammonium acetate buffered at pH 9 and B was acetonitrile. The flow rate was 250 μl/min and the gradient ran from 90% B for 0.5 min to 40% B over 18 min, held at 40% B for a further 3 min, and then returned in 0.5 min to initial conditions (90% B) finally held for 5 min before subsequent analysis. The *m/z* range was 100–1500 and ISpray voltage at 4.2 kV (positive mode) and 3.0 kV (negative mode). Each full MS scan was followed by data dependent MS/MS on the two most intense ions using stepped CID fragmentation mode at 35% normalized collision energy, isolation width of 2 u and activation Q set at 0.250.

## Data analysis and visualization

Peak detection and alignment were performed using MS-DIAL (ver. 3.12) [62]. Peak annotation was done using MS-finder (ver. 3.04) [63] with HMDB, ChEBI, LipidMAPS and Lipid-Blast databases, allowing a level 2.2 of metabolite identification [64,65]. Data were normalized by total ion chromatogram (TIC) and features lower than 2-fold average blank were removed. Each LC-HRMS condition was analyzed separately before concatenation. Data were normalized by auto-scaling before selecting regulated metabolites with more than 2-fold intensity change and a p-value < 0.05 with FDR adjustment as indicate by unpaired t-test using MetaboAnalyst (ver. 4.0) [66]. To account for physiological variations between the different time-tissue combinations, the t-tests were done by comparing the same tissue in infected and uninfected conditions within each time. Uninfected condition was used as control. PCA for quality control was performed with MetaboAnalyst (ver. 4.0). MS/MS similarity metabolic networks with cut-off > 60% were generated with MS-Finder for each LC-HRMS mode-tissue combination.

## dsRNA-mediated RNAi

Templates for dsRNA against AAEL011898 and AAEL001000 were PCR amplified with primers flanked with a T7 promoter (S3 Table) from mosquito cDNA. dsRNA was synthetized with megaScript T7 transcription kit (Thermo Fisher Scientific), extracted in DEPC-treated water and annealed by slowly cooling down from 95˚C. Control dsRNA targeting LacZ was produced [67]. Two- to five-day-old cold-anesthetized female mosquitoes were intrathoracically injected with 69 nl of 3 mg/ml of dsRNA by using Nanoject II injector (Drummond Scientific). Mosquitoes were then maintained at 28˚C with water and 10% sucrose solution before oral infection as detailed above. Cells were seeded at 2 x 10$^5$ per 24-well plate and transfected after 24h with 1 μg of dsRNA by using TransIT-mRNA Transfection kit (Mirusbio). Infection with DENV was done one day post transfection.

## Quantification of DENV genomic RNA

Single mosquitoes or tissues were homogenized in 350μl of TRK lysis buffer (Omega Bio-tek) using a bead Mill homogenizer (FastPrep-24, MP Biomedicals). Total RNA was extracted using E.Z.N.A. Total RNA kit I (Omega Bio-tek) and eluted in 30μl of DEPC-treated water. Genomic RNA (gRNA) was quantified with one-step RT-qPCR using iTaq Universal probe kit (Bio-Rad) and primers and probes targeting the DENV Envelope [68]. The 12.5 μl reaction mix contained 1 μM of forward and reverse primers, 0.125 μM of probe and 4 μl of RNA extract. Quantification was conducted on a CFX96 Touch Real-Time PCR Detection System (Bio-Rad). Thermal profile was 50˚C for 10 min, 95˚C for 1 min and 40 cycles of 95˚C for 10 sec and 60˚C for 15 sec.

An absolute standard curve was generated by amplifying fragments containing the qPCR target using a forward primer tagged with T7 promoter; forward: 5'-CAGGATAAGAGGTT CGTCTG-3' and reverse: 5'-TTGACTCTTGTTTATCCGCT-3', resulting in a 453bp fragment. The fragment was reverse transcribed using MegaScript T7 transcription kit (Ambion) and purified using E.Z.N. A. Total RNA kit I. The total amount of RNA was quantified using a Nanodrop (Thermo Fisher Scientific) to estimate copy number. Ten times serial dilutions were made and used to generate absolute standard equation for gRNA. In each subsequent RT-qPCR plate, five standards were added to adjust for threshold variation between plates. The infection rate was calculated by dividing the number of samples with detectable gRNA over total number of samples.

## Titration

Titration was conducted by plaque assay with BHK-21 cells as described previously [69]. Briefly, 80–90% confluent cells were inoculated with serial 10-fold dilutions of samples for 1h. Cells were then incubated with 1% carboxyl-methyl cellulose (CMC) (Merck) for 5 days, fixed with 4% formaldehyde (Merck) -PBS and stained with 1% crystal violet (Sigma-Aldrich) solution to count plaque forming units (pfu).

## Quantification of gene expression

Total RNA from five mosquitoes was extracted using E.Z.N.A. Total RNA kit I, treated with RapidOut DNA Removal kit (Thermo Fisher Scientific) and reverse transcribed with iScript cDNA Synthesis Kit (Bio-Rad). Gene expression was quantified using iTaq Universal SYBR Green Supermix (Bio-Rad) and primers detailed in S4 Table. *Actin* expression was used for normalization. Quantification was conducted in a CFX96 Touch Real-Time PCR Detection System (Bio-Rad). Thermal profile was 95˚C for 1 min and 40 cycles of 95˚C for 10 sec and 60˚C for 15 sec. Three biological replicates were conducted.

## Statistical analysis

Differences in gRNA copies per infected mosquito were tested on values using unpaired t-test or Mann-Whitney test depending on the normal distribution estimated with D'agostino and Pearson normality test. Differences in percentages were tested using $\chi^2$ test. Tests were performed with GraphPad PRISM software (ver. 6.01).

## AGPAT sequence alignment

Amino acid sequence homology was determined by MEGA X software (ver. 10.0.5), using Maximum Likelihood and bootstrapping. FASTA sequences (S5 Table) were retrieved from ncbi.nlm.nih.gov.

## Supporting information

**S1 Fig. Quantification of DENV infection in Aag2 cells, *A. aegypti* mosquito and midgut.**
(A) DENV gRNA copies in Aag2 cells at 6, 12, 24 and 48h post-infection with DENV at
MOI = 5. Points from 4 repeat and standard errors show geometric mean ± 95% CI. (B) Plaque
titer (plaque forming unit—pfu) in supernatant from Aag2 cells at 6, 12, 24 and 48h post-infec-
tion. Aag2 cells were infected with DENV at MOI = 5 and virus titer was calculated using pla-
que assay. Each point represents one well. (C) DENV gRNA copies per infected mosquitoes
and dissected midguts at 1, 7 and 14 days post-oral infection with $10^7$ pfu/ml. Each point rep-
resents one mosquito or midgut. Bars show geometric mean ± 95% CI. (D) Infection rate in
whole mosquito and midgut at 1, 7 and 14 days post-oral infection. Bars represent
percentages ± s.e.
(TIF)

**S2 Fig. LC-HMRS analytical pipeline.** Liquid Chromatography-High Resolution Mass Spec-
trometry (LC-HRMS) pipeline used to detect polar and nonpolar metabolites.
(TIF)

**S3 Fig. Spectral similarity network from mosquito MS features.** Example of a molecular
spectral network for Ae. aegypti mosquito at 14 days post-infection using MS features detected
with the non-polar LC condition and MS negative mode. Line length represents the MS/MS
score similarity. Ontology for unknown features was determined based on the proximity with
database-identified features.
(TIF)

**S4 Fig. Ion intensity of regulated metabolites in cells, midguts and mosquitoes infected
with DENV and mock.** Normalized ion intensity was calculated after total ion chromatogra-
phy normalization and auto scaling from three biological replicates. Conditions with signifi-
cantly regulated metabolites (p-value <0.05 and |log2 fold change| >1) were indicated with an
asterisk. Only metabolites from the general metabolism (i.e., lipid, carbohydrate, amino acid
and peptide, nucleotide and nucleoside, sialic acid) are shown. †, indicates metabolite anno-
tated by spectral similarity. Carbo., carbohydrate; SA, sialic acid; PL, phospholipid; PE,
phosphatidylethanolamine; PC, phosphatidylcholine; PS, phosphatidylserine; PA, phospha-
tidic acid; PI, phosphatidylinositol; PG, phosphatidylglycerol; LysoPS, lysophosphatidylserine;
LysoPC, lysophosphatidylcholine; LysoPE, lysophosphatidylethanolamine; LysoPG, lysopho-
sphatidylglycerol; LysoPI, lysophosphatidylinositol; SM, Sphingomyelin; DAG, Diacylglycerol;
MAG, Monoacylglycerol; FAHFA, Fatty Acid ester of Hydroxyl Fatty Acid; NAE, N-acyletha-
nolamine; HEA, Heneicosanoic acid; pep., peptides.
(TIF)

**S5 Fig. AGPAT genes are regulated by DENV infection, from Colpitts transcriptomic
data.** *Ae. aegypti* female mosquitoes were inoculated with DENV serotype 2 and collected at 1,
2 and 7 days post inoculation for transcriptomic analysis using microarray. Data from 3 sepa-
rate infections. AGAPAT 1–5 were found significantly regulated by DENV infection. Data
retrieved from (Colpitts et al., 2011, PMID: 21909258).
(TIF)

**S6 Fig. UV-inactivated DENV does not regulate *AGPAT1* expression.** Aag2 cells were
infected with an MOI of 5 of DENV (DENV-WT), UV-inactivated DENV (DENV-UV) or
mock. Cells were analyzed at 24 and 48 hours post-infection (hpi). (A) DENV gRNA copies.
Bars show geometric means ± 95% C.I. (B) *AGPAT1* expression relative to *Actin* level. Bars
show arithmetic means ± s.e.m. (A-B) Each point represents an independent well. ***, p-

value < 0.001 as determined by unpaired t-test.
(TIF)

**S7 Fig. Metabolomic impact of AGPAT1 and AGPAT2 depletion and infection in cells as measured by ion intensity.** Aag2 cells were transfected with dsRNA against *AGPAT1* (dsAGPAT1) or *AGPAT2* (dsAGPAT2) or a dsRNA control (dsControl). 24h later, cells were infected with DENV at MOI of 1. (A) Ion intensity of regulated metabolites in mock cells at 72h post transfection. (B) Ion intensity of regulated metabolites in infected cells at 24 hpi. Normalized ion intensity was calculated after total ion chromatography normalization and auto scaling from three replicates. Conditions with significantly regulated metabolites (p-value <0.05 and |log2 fold change| >1) were indicated with an asterisk. PE, phosphatidylethanolamine; PC, phosphatidylcholine; PS, phosphatidylserine; LPE, lysophosphatidylethanolamine.
(TIF)

**S8 Fig. *AGPAT1* and *2* expression in cells after the other AGPAT depletion.** Aag2 cells were transfected with dsRNA against *AGPAT1* or *2* (dsAGPAT1 or 2). Control cells were transfected with dsRNA control (dsControl). Cells were collected 72h post dsRNA. (A) *AGPAT1* expression in AGPAT2-depleted cells. (B) *AGPAT2* expression in AGPAT1-depleted cells. Bars show mean ± s.e.m from 3 biological replicates. ns, non-significant, as indicated by unpaired t-test.
(TIF)

**S9 Fig. Ethanolamine supplementation partially rescued infection increase upon AGPAT1 depletion.** 24h before infection, Aag2 cells were transfected with dsRNA against AGPAT1 (dsAGPAT1) or with dsRNA control (dsControl) and reared in standard growth media or the same media supplemented with 2mM ethanolamine. Cells were infected with DENV at MOI of 1 and gRNA copy was quantified 48h later. Bars show geometric means ± 95% C.I. Each point represents an independent well. *, p-value < 0.05; **, p-value < 0.01 as determined by unpaired t-test.
(TIF)

**S10 Fig. Metabolomic impact of AGPAT1 and AGPAT2 depletion in uninfected mosquitoes as measured by ion intensity.** Two days post dsRNA injection against *AGPAT1* (dsAGPAT1) or control (dsControl), mosquitoes were orally infected with DENV at $10^7$ pfu/ml. Metabolomic analyses were performed at 2 and 7 dpi. Normalized ion intensity was calculated after total ion chromatography normalization and auto scaling from three replicates. Conditions with significantly regulated metabolites (p-value <0.05 and |log2 fold change| >1) were indicated with an asterisk.
(TIF)

**S11 Fig. Impact of AGPAT1-depletion in mosquitoes on DENV infection rate and gRNA copies.** Mosquitoes were injected with either dsRNA against *AGPAT1* (dsAGPAT1) or dsRNA control (dsControl). Two days post injection, mosquitoes were orally fed with either non-infectious blood or DENV infectious blood. Impact of AGPAT1 depletion on (A) infection rate and (B) DENV gRNA copies at 2 days post oral infection (dpi) with $10^7$ pfu/ml. (C-E) Impact of AGPAT1 depletion on infection rate (C) and DENV gRNA copies at 2 (D) and 7 (E) dpi with $10^6$ pfu/ml. Bars indicate percentage ± s.e. (A, C) or geometric means ± 95% C.I. (B, D, E) with each dot representing one mosquito. *, p-value < 0.05 as indicated by Mann-Whitney test.
(TIF)

**S1 Table. Identification of mosquito specific metabolites by spectral similarity.**
(DOCX)

**S2 Table. Metabolites detected from cell, midgut and whole mosquito with differential regulation upon DENV infection.** The three tabs contain compound detected on cell, midgut and mosquito with the following information: ionization mode (positive and negative), phase detection (polar and non-polar), the mass average m/z between replicate, the retention time average Rt in minutes between replicates, the MS/MS spectrum fragmentation and intensity, adducts [M+H]+ and [M-H]-, metabolites of importance in Fig 1C, regulated compound with abundance between DENV-infected and uninfected samples (p-value < 0.05 and |log2 fold change| ≥ 1) and annotation classes with the first 3 ranks of identification by MS-Finder.
(XLSX)

**S3 Table. Primers for dsRNA.**
(DOCX)

**S4 Table. Primers for Real-Time qPCR.**
(DOCX)

**S5 Table. AGPAT FASTA protein sequences.**
(DOCX)

**S6 Table. Metabolites detected from uninfected cell, after AGPAT1 or AGPAT2 depletion.**
The table contains compound detected on cell with the following information: ionization mode (positive and negative), the mass average m/z between replicate, the retention time average Rt in minutes between replicates, the MS/MS spectrum fragmentation and intensity, adducts [M+H]+ and [M-H]-, metabolites of importance in Fig 3C, regulated compound with abundance between dsAGPAT1 or 2 and dsControl samples (p-value < 0.05, |log2 fold change| ≥ 1) and annotation classes with the first 3 ranks of identification by MS-Finder.
(XLSX)

**S7 Table. Metabolites detected from DENV-infected cell at 48 hpi, after AGPAT1 or AGPAT2 depletion.** The table contains compound detected on cell with the following information: ionization mode (positive and negative), the mass average m/z between replicate, the retention time average Rt in minutes between replicates, the MS/MS spectrum fragmentation and intensity, adducts [M+H]+ and [M-H]-, metabolites of importance in Fig 3E, regulated compound with abundance between dsAGPAT1 or 2 and dsControl samples (p-value < 0.05, |log2 fold change| ≥ 1) and annotation classes with the first 3 ranks of identification by MS-Finder.
(XLSX)

**S8 Table. Metabolites detected from uninfected mosquito after AGPAT1 depletion.** The two tabs contain compound detected on cell after 2 or 7 days post blood feeding with the following information: ionization mode (positive and negative), the mass average m/z between replicate, the retention time average Rt in minutes between replicates, the MS/MS spectrum fragmentation and intensity, adducts [M+H]+ and [M-H]-, metabolites of importance in Fig 4B, regulated compound with abundance between dsAGPAT1 and dsControl samples (p-value < 0.05, |log2 fold change| ≥ 1) and annotation classes with the first 3 ranks of identification by MS-Finder.
(XLSX)

**S9 Table. Metabolites detected from DENV-infected mosquito after dsAGPAT1 depletion or dsControl.** The table contains compound detected on cell with the following information:

ionization mode (positive and negative), the mass average m/z between replicate, the retention time average Rt in minutes between replicates, the MS/MS spectrum fragmentation and intensity, adducts [M+H]+ and [M-H]-, metabolites of importance in Fig 4E, regulated compound with abundance between dsAGPAT1 and dsControl samples (p-value < 0.05, |log2 fold change| ≥ 1) and annotation classes with the first 3 ranks of identification by MS-Finder. (XLSX)

**S10 Table. Metabolites detected from DENV-infected or uninfected mosquito after AGPAT1 depletion.** The table contains compound detected on cell with the following information: ionization mode (positive and negative), the mass average m/z between replicate, the retention time average Rt in minutes between replicates, the MS/MS spectrum fragmentation and intensity, adducts [M+H]+ and [M-H]-, metabolites of importance in Fig 4F, regulated compound in dsAGPAT1 condition only, regulated compound with abundance between DENV-infected and Mock samples in dsControl or dsAGPAT1 condition (p-value < 0.05, |log2 fold change| ≥ 1) and annotation classes with the first 3 ranks of identification by MS-Finder. (XLSX)

## Acknowledgments

We are grateful to Dr. Mariano Garcia-Blanco for constant support and the Pompon/Garcia-Blanco's team for comments on a previous version. We thank Dr. Eng Eong Ooi for providing the ST virus. We also thank the Mass spectrometry platform, Université de Toulouse, ICT, UPS, Toulouse, France.

## Author Contributions

**Conceptualization:** Dorothée Missé, Guillaume Marti, Julien Pompon.

**Data curation:** Thomas Vial.

**Formal analysis:** Thomas Vial, Guillaume Marti, Julien Pompon.

**Funding acquisition:** Eric Deharo, Julien Pompon.

**Investigation:** Thomas Vial, Wei-Lian Tan, Benjamin Wong Wei Xiang, Guillaume Marti.

**Methodology:** Thomas Vial, Guillaume Marti, Julien Pompon.

**Supervision:** Eric Deharo, Guillaume Marti, Julien Pompon.

**Visualization:** Thomas Vial.

**Writing – original draft:** Thomas Vial, Guillaume Marti, Julien Pompon.

**Writing – review & editing:** Thomas Vial, Dorothée Missé, Eric Deharo, Guillaume Marti, Julien Pompon.

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
