## [Decision Letter · Decision Letter 0]

16 Jul 2019

Dear Dr Pompon,

Thank you very much for submitting your manuscript "Dengue virus inhibits AGPAT1 to alter phospholipids and enhance infection in Aedes aegypti" (PPATHOGENS-D-19-00992) for review by PLOS Pathogens. Your manuscript was fully evaluated at the editorial level and by independent peer reviewers. The reviewers appreciated the attention to an important problem, but raised some substantial concerns about the manuscript as it currently stands. These issues must be addressed before we would be willing to consider a revised version of your study. We cannot, of course, promise publication at that time.

We therefore ask you to modify the manuscript according to the review recommendations before we can consider your manuscript for acceptance. Your revisions should address the specific points made by each reviewer.

(1) A letter containing a detailed list of your responses to the review comments and a description of the changes you have made in the manuscript. Please note while forming your response, if your article is accepted, you may have the opportunity to make the peer review history publicly available. The record will include editor decision letters (with reviews) and your responses to reviewer comments. If eligible, we will contact you to opt in or out.

(2) Two versions of the manuscript: one with either highlights or tracked changes denoting where the text has been changed; the other a clean version (uploaded as the manuscript file).

Additionally, to enhance the reproducibility of your results, PLOS recommends that you deposit your laboratory protocols in protocols.io, where a protocol can be assigned its own identifier (DOI) such that it can be cited independently in the future. For instructions see http://journals.plos.org/plospathogens/s/submission-guidelines#loc-materials-and-methods

We hope to receive your revised manuscript within 60 days. If you anticipate any delay in its return, we ask that you let us know the expected resubmission date by replying to this email. Revised manuscripts received beyond 60 days may require evaluation and peer review similar to that applied to newly submitted manuscripts.

There is additional documentation related to this decision letter. To access the file(s), please click the link below. You may also login to the system and click the 'View Attachments' link in the Action column.

[LINK]

Sincerely,

Glenn Randall

Associate Editor

PLOS Pathogens

Ana Fernandez-Sesma

Section Editor

PLOS Pathogens

Kasturi Haldar

Editor-in-Chief

PLOS Pathogens

orcid.org/0000-0001-5065-158X

Grant McFadden

Editor-in-Chief

PLOS Pathogens

orcid.org/0000-0002-2556-3526

In particular, please note the need to establish causative and not correlative links, in addition to providing much more mechanistic detail as to how DENV modulates AGPAT1.

Reviewer's Responses to Questions

**Part I - Summary**

Reviewer #1: The authors perform metabolomic analysis on dengue infected cells and mosquitoes, which broadly agrees with previously published metabolomic studies on dengue infection of mosquitoes. They find that dengue infection induces substantial changes to various aminophospholipid species. Further, they find the dengue infection substantially reduces the expression of AGPAT, a key enzyme in phospholipid synthesis. siRNA-mediated targeting of AGPAT moderately increases viral replication in cell culture and in mosquitoes. Further, they find that siRNA-mediated depletion of AGPAT phenocopies aspects of the metabolic changes induced by dengue infection. The identification of AGPAT1 as a restriction factor whose downregulation could be contributing to dengue-induced metabolic remodeling has implications for the field’s understanding of dengue-induced metabolic remodeling, but as indicated below the authors over-interpret a number of correlations implying causality, which, when coupled to a lack of mechanistic details of how dengue targets AGPAT1 or how AGPAT1 contributes to dengue infection limit the impact of the current work.

Reviewer #2: This study aims to understand how dengue virus (DENV) modulates mosquito metabolome to enhance virus production. Using high-resolution mass spectrometry, the authors identified changes in aminophospholipid (aminoPL) concentrations during the course of DENV infection in vitro and in vivo. They showed that the expression of 1-Acylglycerol-3-Phosphate O-Acyltransferase 1 (AGPAT1), a rate-limiting enzyme of phospholipid biogenesis, is decreased during DENV infection, whereas AGPAT1 knockdown leads to a modest increase in DENV production late in infection. This is a significant, but highly descriptive study. There is little mechanism as to how DENV utilizes a select group of phospholipids (PL) to enhance virus production. The study does not address how DENV infection causes AGPTA1 inhibition or the relationship between AGPAT1 inhibition and an increase in aminoPL concentration. It is unclear how in aminoPLs contribute to DENV infectivity in mosquitoes. Finally, if AGPTA1 plays a crucial role in DENV infectivity, its silencing should lead to a marked increase in DENV load. However, this is not the case in the current study.

Reviewer #3: In the manuscript entitled “Dengue virus inhibits AGPAT1 to alter phospholipids and enhance infection in Aedes aegypti,” Vial et al study the metabolome of Aedes Aegypti mosquitos over time after dengue infection. While other metabolomics studies of the mosquito midgut following Dengue infection have been published (Chotiwan, PLoS Pathogens 2018), this one is more comprehensive including Aedes Aegypti cells, midguts, and whole mosquitos. Furthermore, this study goes beyond the previous one by examining the role of a host lipid synthesis enzyme in this process. The strengths of this manuscript include the provision of a comprehensive database reflecting metabolic changes during dengue infection of mosquitos and the uncovering of a novel pathogenesis mechanism for Dengue in which a mosquito gene that inhibits viral replication is downregulated by the virus during infection. Weaknesses include the presentation of the findings and statistical analysis of the data.

**Part II – Major Issues: Key Experiments Required for Acceptance**

Reviewer #1: Throughout the manuscript, the authors conclude that inhibition of AGPAT1 promotes infection by increasing aminophospholipids. While their data show a correlation between aminophospholipids levels, and dengue infection, a causal link between aminophospholipid levels and infection is never established. Are downstream aminophospholipid producing enzymes important for dengue infection? Does supplementation with specific aminophospholipids impact infection, e.g. in vitro?

The authors claim, e.g. in the title, that dengue inhibits AGPAT1, this suggests inhibition of the protein. What the authors actually show is that AGPAT1 expression levels are down-regulated. What mediates this down regulation? Is live virus necessary for AGPAT1 down-regulation? Does UV-inactivated virus down-regulate AGPAT1 expression?

Does AGPAT1 downregulation impact the phospholipid constituents of the virion? Or is the mosquito’s innate immune response to viral infection affected? In vitro, what aspects of the viral life cycle are impacted?

Reviewer #2: Fig. 1 addresses the impact of DENV infection on Ae. Aegypti metabolome. In Fig. 1B, the Venn diagrams and related triangles need to be explained. In addition, there are no controls for the heat maps in Fig. 1C, making it hard to judge the data.

Fig. 2C-E show a decrease in AGPTA1 mRNA levels, during the course of DENV infection. These data need to be validated with AGPTA1 protein expression during DENV infection.

Fig. 3 addresses the impact of AGPAT silencing on aminoPL levels and DENV replication in mosquito cells. First, it will be helpful to show the control PL samples for the heat maps in Fig. 3C and 3E. It is hard to compare the PL heat maps, from mock and DENV-infected cells, when they show different PL species. In Fig. 3D, it is also hard to interpret the DENV infection data based on the heat maps in Fig. 3C and 3E.There is only a modest increase in DENV production after AGPAT1 knockdown.

Fig. 4 addresses the impact of AGPAT1 silencing on DENV replication and aminoPL consumption in mosquitoes. Please, show the heat map for control siRNA-treated mosquitoes in Fig. 4B. In Fig. 4CD, contrary to the authors assertion, AGPAT1 knockdown in mosquitoes shows no significant impact on DENV infectivity at day 2 or 7 postinfection. The DENV infection rate at day 7 (Fig. 4C) is clearly not impacted by AGPAT1 knockdown. There might be a slight increase in DENV genome replication at day 7, but is there a significant change in DENV load in these mosquitoes?

Reviewer #3: 1) Figure 2: AGPAT1 expression is decreased in cells and whole mosquitos. Given that the virus disseminates from the midgut and that the decrease in AGPAT1 expression does not explain the observed increase in aminophospholipids in the midgut early in infection, the authors should also measure AGPAT1 and 2 expression in the mosquito midgut.

2) Figure 3D: the control should have as many data points as the test samples.

**Part III – Minor Issues: Editorial and Data Presentation Modifications**

Reviewer #1: A thorough metabolomics analysis of dengue infection in mosquitoes was previously published in Plos Pathogens in early 2018 (PMID: 29447265), which has somewhat similar findings. Similarities and potential differences should be addressed in the discussion.

In the abstract, the authors claim the dengue infection increases AGPAT1, when all their data indicates that dengue markedly reduces it. This is likely just a typo, but should clearly be fixed as it is a central conclusion.

In Figure 1, its unclear which of the metabolites are significant. The figures would be easier to interpret with an added scheme to enable readers to quickly assess significance of specific changes.

In Figure 3C and 3E, the data should not be shown as a heat map. They should be plotted so that fold changes and data variance/errors can be easily assessed.

Reviewer #2: The manuscript needs editing.

Reviewer #3: 1) Abstract: While the abstract length is limited, it is critical to spell out AGPAT and its function in one sentence so readers are not left in the dark. An indication of which aminophospholipids the study is focused on might also be helpful.

2) Line 22: Is this a typographical error? Figure 2 shows that Dengue infection decreases expression of AGPAT1 rather than the opposite.

3) Line 39, sentence beginning with “however:” This sentence is confusing and should be rewritten.

4) Line 147: “PL were the most regulated.” This is a vague statement. Do the authors mean that the largest changes in abundance were seen for these metabolites over the course of the infection?

5) Lines 151-189: In this section, the authors describe in detail the changes in different types of lipids without adding any insight into processes that are at work. These changes are already apparent in Figure 1 and could also be gleaned from supplementary datasets. I would suggest shortening this section and only describing in prose changes that provide insight into the possible underlying mechanisms.

6) Lines 194-5: Please describe what “the reaction” is.

7) Lines 208-9 and Figure 2: Upregulation of AGPAT2 is not convincing and not significant. I would recommend leaving this statement out.

8) Figure 2: AGPAT1 expression is decreased in cells and whole mosquitos. Given that the virus disseminates from the midgut and that the decrease in AGPAT1 expression does not explain the observed increase in aminophospholipids in the midgut early in infection, the authors should also measure AGPAT1 and 2 expression in the mosquito midgut.

9) Figure 3D: The increase in PFU is quite small, suggesting that this may be a minor contributor. More importantly:

An unpaired t-test is used to assess statistical significance. A multiple comparisons test such as an ANOVA would be a more appropriate statistical test to apply here.

10) Methods: The authors state that they used an unpaired t-test to analyze the metabolomics data. The authors should specify which condition was used as the “control.” In addition, for metabolomics time courses such as that shown in Figure 1C, a multiple comparisons test should be used to assess statistical significance rather than an unpaired t test.

11) Methods: Please specify when replicates are experimental and when they are technical replicates rather than describing as “repeats.”

PLOS authors have the option to publish the peer review history of their article (what does this mean?). If published, this will include your full peer review and any attached files.

Reviewer #1: No

Reviewer #2: No

Reviewer #3: No

---

## [Editor Report · Decision Letter 1]

24 Oct 2019

Dear Dr Pompon,

Thank you very much for submitting your manuscript "Dengue virus reduces AGPAT1 expression to alter phospholipids and enhance infection in Aedes aegypti" (PPATHOGENS-D-19-00992R1) for review by PLOS Pathogens. The editors have reviewed your revision and would like you  to experimentally address some of the points made by Reviewer #1. In particular, is the issue of whether phopsphatidic acid (or its downstream products) is inhibitory to DENV replication in vitro (Aag cells) both in wild type infection and the silencing experiments. We view this as important for two reasons. (i) It further controls the silencing experiment for off-target effects and directly implicates the lipid pathway. (ii) It asks the important question as to what happens when DENV can't downregulate phosphatidic acid. Silencing AGPAT1 likely produces a minor 2-fold phenotype on DENV replication because DENV infection is already decreasing its expression level.  This experiment can be done via simple media addition of PA over a range of concentrations or over-expression of AGPAT1 if necessary. qRT-PCR and titering would give some insight into life cycle stage (controlling for cell viability). This should be a relatively simple experiment. An additional in vivo experiment is unnecessary, as it is difficult to adequately control.

These issues must be addressed before we would be willing to consider a revised version of your study. We cannot, of course, promise publication at that time.

We therefore ask you to modify the manuscript according to the review recommendations before we can consider your manuscript for acceptance. Your revisions should address the specific points made by each reviewer.

(1) A letter containing a detailed list of your responses to the review comments and a description of the changes you have made in the manuscript. Please note while forming your response, if your article is accepted, you may have the opportunity to make the peer review history publicly available. The record will include editor decision letters (with reviews) and your responses to reviewer comments. If eligible, we will contact you to opt in or out.

(2) Two versions of the manuscript: one with either highlights or tracked changes denoting where the text has been changed; the other a clean version (uploaded as the manuscript file).

Additionally, to enhance the reproducibility of your results, PLOS recommends that you deposit your laboratory protocols in protocols.io, where a protocol can be assigned its own identifier (DOI) such that it can be cited independently in the future. For instructions see http://journals.plos.org/plospathogens/s/submission-guidelines#loc-materials-and-methods

We hope to receive your revised manuscript within 60 days. If you anticipate any delay in its return, we ask that you let us know the expected resubmission date by replying to this email. Revised manuscripts received beyond 60 days may require evaluation and peer review similar to that applied to newly submitted manuscripts.

[LINK]

Sincerely,

Glenn Randall

Associate Editor

PLOS Pathogens

Ana Fernandez-Sesma

Section Editor

PLOS Pathogens

Kasturi Haldar

Editor-in-Chief

PLOS Pathogens

orcid.org/0000-0001-5065-158X

Grant McFadden

Editor-in-Chief

PLOS Pathogens

orcid.org/0000-0002-2556-3526

---

## [Editor Report · Decision Letter 2]

7 Nov 2019

Dear Dr Pompon,

We are pleased to inform that your manuscript, "Dengue virus reduces AGPAT1 expression to alter phospholipids and enhance infection in Aedes aegypti", has been editorially accepted for publication at PLOS Pathogens. 

Before your manuscript can be formally accepted and sent to production, you will need to complete our formatting changes, which you will receive by email within a week. Please note that your manuscript will not be scheduled for publication until you have made the required changes.

IMPORTANT NOTES

(1) Please note, once your paper is accepted, an uncorrected proof of your manuscript will be published online ahead of the final version, unless you’ve already opted out via the online submission form. If, for any reason, you do not want an earlier version of your manuscript published online or are unsure if you have already indicated as such, please let the journal staff know immediately at plospathogens@plos.org.

(2) Copyediting and Proofreading: The corresponding author will receive a typeset proof for review, to ensure errors have not been introduced during production. Please review the PDF proof of your manuscript carefully, as this is the last chance to correct any errors. Please note that major changes, or those which affect the scientific understanding of the work, will likely cause delays to the publication date of your manuscript. 

(3) Appropriate Figure Files: Please remove all name and figure # text from your figure files. Please also take this time to check that your figures are of high resolution, which will improve the readbility of your figures and help expedite your manuscript's publication. Please note that figures must have been originally created at 300dpi or higher. Do not manually increase the resolution of your files. For instructions on how to properly obtain high quality images, please review our Figure Guidelines, with examples at: http://journals.plos.org/plospathogens/s/figures.

(4) Striking Image: Please upload a striking still image to accompany your article if one is available (you can include a new image or an existing one from within your manuscript). Should your paper be accepted, this image will be considered for our monthly issue image and may also appear on our website to feature your article. Please upload this as a separate file, selecting "striking image" as the file type upon upload. Please also include a separate "Other" file with a caption, including credits and any potential copyright information. Please do not include the caption in the main article file. If your image is from someone other than yourself, please ensure that the artist has read and agreed to the terms and conditions of the Creative Commons Attribution License at http://journals.plos.org/plospathogens/s/content-license. Please note that PLOS cannot publish copyrighted images.

(5) Press Release or Related Media: If your institution or institutions have a press office, please notify them about your upcoming paper at this point, to enable them to help maximize its impact. If they will be preparing press materials for this manuscript, please inform our press team in advance at plospathogens@plos.org as soon as possible. We ask that you contact us within one week to plan ahead of our fast Production schedule. If you need to know your paper's publication date for related media purposes, you must coordinate with our press team, and your manuscript will remain under a strict press embargo until the publication date and time. This means an early version of your manuscript will not be published ahead of your final version. 

(6)  PLOS requires an ORCID iD for all corresponding authors on papers submitted after December 6th, 2016. Please ensure that you have an ORCID iD and that it is validated in Editorial Manager.  To do this, go to ‘Update my Information’ (in the upper left-hand corner of the main menu), and click on the Fetch/Validate link next to the ORCID field.  This will take you to the ORCID site and allow you to create a new iD or authenticate a pre-existing iD in Editorial Manager

(7) Update your Profile Information: Now that your manuscript has been provisionally accepted, please log into Editorial Manager and update your profile, if needed. Go to https://www.editorialmanager.com/ppathogens, log in, and click on the "Update My Information" link at the top of the page. Please update your user information to ensure an efficient production and billing process. 

(8) LaTeX users only: Our staff will ask you to upload a TEX file in addition to the PDF before the paper can be sent to typesetting, so please carefully review our Latex Guidelines http://journals.plos.org/plospathogens/s/latex in the meantime.

(9) If you have associated protocols in protocols.io, please ensure that you make them public before publication to guarantee immediate access to the methodological details.

Best regards,

Glenn Randall

Associate Editor

PLOS Pathogens

Ana Fernandez-Sesma

Section Editor

PLOS Pathogens

Kasturi Haldar

Editor-in-Chief

PLOS Pathogens

orcid.org/0000-0001-5065-158X

Grant McFadden

Editor-in-Chief

PLOS Pathogens

orcid.org/0000-0002-2556-3526
---

## [Editor Report · Acceptance letter]

2 Dec 2019

Dear Dr Pompon,

We are delighted to inform you that your manuscript, "Dengue virus reduces AGPAT1 expression to alter phospholipids and enhance infection in Aedes aegypti," has been formally accepted for publication in PLOS Pathogens.

Best regards,

Kasturi Haldar

Editor-in-Chief

PLOS Pathogens

orcid.org/0000-0001-5065-158X

Grant McFadden

Editor-in-Chief

PLOS Pathogens

orcid.org/0000-0002-2556-3526